# Knee Pain from Osteoarthritis: Pathogenesis, Risk Factors, and Recent Evidence on Physical Therapy Interventions

**DOI:** 10.3390/jcm11123252

**Published:** 2022-06-07

**Authors:** Jean-Philippe Berteau

**Affiliations:** 1Department of Physical Therapy, City University of New York-College of Staten Island, Staten Island, NY 10314, USA; jean.berteau@csi.cuny.edu; 2New York Center for Biomedical Engineering, City University of New York-City College of New York, New York, NY 10016, USA; 3Nanoscience Initiative, Advanced Science Research Center, City University of New York, New York, NY 10031, USA

**Keywords:** knee pain, osteoarthritis, physical therapy, orthopedics, rehabilitation

## Abstract

For patients presenting knee pain coming from osteoarthritis (OA), non-pharmacological conservative treatments (e.g., physical therapy interventions) are among the first methods in orthopedics and rehabilitation to prevent OA progression and avoid knee surgery. However, the best strategy for each patient is difficult to establish, because knee OA’s exact causes of progression are not entirely understood. This narrative review presents (i) the most recent update on the pathogenesis of knee OA with the risk factors for developing OA and (ii) the most recent evidence for reducing knee pain with physical therapy intervention such as Diathermy, Exercise therapy, Ultrasounds, Knee Brace, and Electrical stimulation. In addition, we calculated the relative risk reduction in pain perception for each intervention. Our results show that only Brace interventions always reached the minimum for clinical efficiency, making the intervention significant and valuable for the patients regarding their Quality of Life. In addition, more than half of the Exercise and Diathermy interventions reached the minimum for clinical efficiency regarding pain level. This literature review helps clinicians to make evidence-based decisions for reducing knee pain and treating people living with knee OA to prevent knee replacement.

## 1. Introduction

Knee osteoarthritis (OA) is a degenerative joint disease and the most common reason for knee joint replacements in the US, with 4.7 million individuals having undergone surgery in 2010 [1] with an associated cost of USD 29,488 Per surgery [2]. The high prevalence of knee OA manifests in enormous societal and personal expenses and urges to prevent OA progression to avoid surgery. Knowledge of a patient’s risk factors helps inform them of their prognosis, and clinicians must adapt the trajectory of a patient’s treatment progress to their needs to maintain functionality. Along with using medicine (e.g., acetaminophen, nonsteroidal anti-inflammatory drugs, and duloxetine), the patient’s treatment starts with physical therapy (e.g., diathermy, exercise therapy, ultrasounds, knee brace, and electrical stimulation). However, the best strategy for each patient is difficult to establish because knee OA’s exact causes of progression are not entirely understood [3]. To help clinicians treat people living with knee OA and prevent knee replacement, this narrative review presents (i) the most recent updates on the pathogenesis of OA with the risk factors for developing OA and (ii) the most recent evidence for physical therapy.

## 2. Pathogenesis and Risk Factors

### 2.1. Pathogenesis

First, OA has been depicted as the result of progressive articular cartilage degradation. Indeed, although the cartilage can prevent biomechanical damage caused by severe loading, patients with OA hinder attempts at repair and result in disrupted cartilage homeostasis [4]. For instance, cartilage cells’ (i.e., chondrocytes’) compositional and structural alterations—such as hypertrophy due to aging or oxidative stress—trigger the production of catabolic factors, enhancing cartilage debilitation. These catabolic factors such as cytokines, chemokines, and proteolytic enzymes—cytokines (e.g., IL-6, IL-8), chemokines (e.g., RANTES, IP-10), metalloproteases (MMP1, MMP3), and heat-shock proteins (e.g., HSPA1A)—have been identified as quantifiable biomarkers for predicting the onset and progression of knee OA. Therefore, for decades, cartilage degradation resulting from the extracellular matrix’s destruction has been depicted as one of the significant biological starters of the OA pathological process.

When the pathological process of knee OA is triggered by catabolic factors production, the articular cartilage of the knee starts to degrade, making it unable to fully absorb physiological and physical forces. This induces associated joint conformational changes that compensate for the loss of articular cartilage, showing that OA is an active repair process [5]. These changes include subchondral bone (SB) sclerosis—thickening and hardening—and the formation of bone cysts and marginal osteophytes (coming from bone remodeling). Thus, all these SB alterations cause the joint space to narrow [6], enhancing OA progression. Ultimately, OA affects the whole joint due to synovial inflammation and fibrosis of the joint capsule [7], which cause loss of range of motion/stiffness, tenderness, and pain. This pathological process has been described in the shape of a vicious circle of OA when one event triggers the other [6].

Regarding pain in OA disease, it involves complex peripheral and central mechanisms. For instance, nerve sensitizations are significant characteristics of pain transmission in OA patients that may contribute to the discordance between pain and joint pathology [8]. Since hyaline cartilage is not innervated, the pain comes from the synovium, subchondral bone, and periosteum, which are innervated by small-diameter nociceptive neurons. The nociceptive stimuli are generated by tissue damage during joint degradation. Previous studies showed pain had been associated with many structural factors, including bone marrow lesions, synovial thickening (synovitis), and knee effusion [9]. The inflammatory mediators produced by the synovium and chondrocytes increase the excitation of the nociceptive neurons, creating an amplified painful response [10]. Recent evidence shows that SB is also a starter of the OA vicious circle. Indeed, OA is also looked at as joint failure caused by abnormal joint loading instead of a disease of cartilage degradation [11]. Specifically, changes in the SB—which can trigger pain through nociceptive stimuli—predispose the cartilage to further damage from wear and tear, as the SB is less able to absorb forces/load placed on the joint [12,13,14,15,16,17]. It is believed that changes to the mechanical properties of SB occur during remodeling, such as bone hardening [16], which induces increased stiffness that precedes and contributes to cartilage loss. Indeed, it has been shown that changes in gene expression of SB precede cartilage degeneration and alter the activity of catabolic factors by chondrocytes contributing to the degeneration of cartilage [15]. Thus, alterations of physiological cross-talk between SB and cartilage [18] are considered the primary trigger of the OA pathological process. We can describe OA as a self-sustaining vicious cycle (adapted from [6]) where each step in the process influences and amplifies each other (Figure 1).

### 2.2. Risk Factors

The development of the vicious cycle of OA is complex and is caused by both modifiable and nonmodifiable risk factors. Indeed, no one risk factor contributes to the increase in the disease process; rather, the involvement of risk factors together such as age, gender, ethnicity, genetic predisposition, hormonal factors, and bone density. In addition, biomechanical factors—caused by sports, the workplace, joint misalignment, and obesity—contribute to joint injuries leading to OA [6,10,19,20].

#### 2.2.1. Age

There is an exponential increase in OA in adults over 50 years old. Indeed, the aging process results in the chondrocytes’ inability to produce proteoglycans to maintain the cartilage matrix—which gives the cartilage its compressive strength—and the failure to maintain homeostasis [10]. Thus, the tissue is less likely to heal when stressed, causing articular cartilage degeneration, leading to OA. However, it cannot be a purely age-related joint wear and tear disease because not all joints are equally affected, and OA changes can develop without aging [10]. For instance, although OA rarely occurs in youth, people with sports injuries younger than 30 years old are at increased risk [21].

#### 2.2.2. Obesity

Individuals living with obesity have a 66% chance of developing symptomatic knee OA compared to a 45% chance of developing OA for people with a conventional weight. In addition, the Framingham OA study [22] shows that women who lost about 5 kg—2 units of body mass index—reduced their risk of knee OA by a half [21]. Obesity increases the risk of developing OA through systemic and biomechanical factors. For instance, obesity alters (i) metabolism and joint inflammation that contributes to OA in non-weight-bearing joints such as the hands and (ii) biomechanical loading on weight-bearing joints such as the knee or the hip. Regarding the biomechanical factors-based on the multiplier effect of lever arms outside the body’s central axis, a force of three to six times the body weight is exerted across the knee during a single-leg stance in walking. Thus, in an individual living with obesity, the increase in weight multiplies the force across the knee during walking, putting the joint’s tissue at high risk of damage [23]. However, whether the weight has a limited effect on the progression of knee OA to moderately misaligned knees (2–7 degrees), knees that were severely misaligned would lead to an OA joint regardless of the weight added to it [24]. In addition, the correlation between obese patients and OA is further strengthened by the development of adipose tissue that secretes adipokines. Indeed, this biologically active substance contributes to joint inflammation that alters cartilage homeostasis, making them more susceptible to OA [24].

#### 2.2.3. Biomechanical Load

Biomechanical overload of a joint through activities requiring repetitive and excessive joint loading, such as knee bending, is associated with knee OA. Indeed, cartilage loss is a mechanically mediated process that is more likely to occur in areas of high stress [11], where an increased expression of cytokines, chemokines, and proteolytic enzymes—PICs and MMPs—was found in response to high fluid shear stress [25]. Similarly, an increase in pressure on the posterior horn of the meniscus during occupational activities with deep flexion loading initiates the degenerative process in the joint [26]. For instance, high-impact sports activities, such as hockey, football, and soccer, lead to undue stress on joints and increase knee OA risk in adults [20,27]. While deep squatting has been shown to increase compressive and posterior shear forces on the knee, 7 and 5 times the body weight, respectively, it is not yet proven that it leads to OA [28].

Joint malalignment—changes in joint geometry—decreases the joint’s ability to adapt to its biomechanical environment, contributing to cartilage or bone tissue damage. For instance, varus knee malalignment and dynamic knee adduction moments have been found to cause medial compartment knee OA due to the increase in mechanical stress on the medial compartment of the knee; the reverse is valid for a valgus knee alignment [29,30,31]. In addition, leg length discrepancies lead to asymmetrical joint mechanics during weight-bearing activities, contributing to the development of hip OA. To compensate for the differences, an individual may increase knee flexion or hip adduction of the longer limb during stance, increasing the force at those joints [20,28].

The biomechanical load can also lead to sports-related joint injuries, a risk factor for OA. For instance, the lack of a functionally standard ACL or meniscus changes the static and dynamic loading of the knee, generating increased forces on the cartilage and SB, leading to OA [32]. Indeed, among Swedish soccer players, radiographic OA—14 years after injuring the ACL—was present in 41% of injured knees compared to 4% in uninjured knees (no difference if there was surgical intervention). In relation to this, in long-term follow-up studies of young athletes with meniscus surgery, more than 50% had OA and associated pain and functional impairment [28]. Thus, OA is increasingly thought of as joint failure driven by abnormal joint loading rather than a discrete disease entity. It is more and more considered a primarily mechanical problem, where the risk factors are all found to affect the biomechanical loading of the joint, contributing to the disease progression.

## 3. Physical Therapy Interventions

The number of treatments for OA is extensive [26,33,34,35], but the effectiveness behind many of them is sporadic. Regarding the Physical Therapy Interventions, one of the best ways to measure the efficiency is the WOMAC (Western Ontario and McMaster Universities Osteoarthritis Index) score. The minimum clinical efficiency associated is roughly a decrease of 20% for each WOMAC sub-scales [36]. Here, we performed a literature review on the primary physical therapy interventions used by clinicians specializing in knee OA rehabilitation, such as diathermy, exercise therapy, ultrasound knee brace, and electrical stimulation. We used the WOMAC pain score as a primary outcome. We calculated the Relative Risk (RR) reduction following the 2005 University of Oxford guidelines, where a RR > 1 indicates that the treatment increased the risk of the outcome according to the following formula and reported as a percentage of increase in the tables.
(1)$$\mathrm{Relative}\;\mathrm{Risk}=\frac{\mathrm{Risk}\;\mathrm{of}\;\mathrm{outcome}\;\mathrm{in}\;\mathrm{the}\;\mathrm{treatment}\;\mathrm{group}}{\mathrm{Risk}\;\mathrm{of}\;\mathrm{out}\;\mathrm{come}\;\mathrm{in}\;\mathrm{the}\;\mathrm{control}\;\mathrm{group}\;},$$

### 3.1. Diathermy

Diathermy, or heat therapy, has been used as a treatment method for varying musculoskeletal issues. The rationale behind diathermy use lies within its ability to increase the temperature of the underlying tissue. An increase in tissue temperature can induce vasodilatation, increase cellular activity, increase pain threshold, increase soft tissue extensibility and reduce muscle spasms [37]. Two forms of diathermy often used are short-wave diathermy (SWD) and microwave diathermy (MD). SWD uses high-frequency electromagnetic energy to generate heat on a particular tissue in a pulsed or continuous wave [38]. MD uses microwaves to generate heat on superficial tissues, as their lower-frequency waves do not penetrate deep muscle [39]. For microwave diathermy, the mechanism of action is believed to increase local blood flow and allow nutrients and oxygen to be delivered to promote tissue repair [39,40]. Indeed, the increased capillary permeability induced by the deep microwave diathermy allows macrophages and granulocytes to reach the affected area, thus removing toxins and necrotic debris. Hyperthermia can interfere with enzymes involved in the inflammatory process, and local microwave diathermy may induce the expression of heat shock proteins, which are essential for proper protein folding and the removal of cellular waste material [39,40].

In our literature review (Table 1), the analysis of interventions was split into classes of diathermy treatments versus control (sham) diathermy and different kinds of diathermy interventions (continuous vs. pulse and superficial vs. deep diathermy) against each other. Deep microwave diathermy has been shown to reduce synovial thickness—a prognostic marker of cartilage loss—in patients with knee OA, which in turn assists in decreasing pain associated with synovitis and the progression of cartilage loss [39]. Multiple studies found a difference in pain scores for patients treated with diathermy; differences ranged from an 8% to 45% decrease in WOMAC scores for various diathermy treatments. Both diathermy therapy and sham therapy showed an improvement in WOMAC scores. There was no significant difference between short wave diathermy and sham diathermy treatments. However, deep microwave diathermy proved to be the most efficient intervention in treating knee OA. Our conclusions depict that the most efficient treatment was the deep microwave diathermy delivered at 434 MHz for 30 min, five times a week.

### 3.2. Exercise Therapy

Quadriceps and hamstring muscle weaknesses worsen knee OA effects by decreasing dynamic muscle actions, losing joint motion [44], and decreasing neuromuscular (proprioceptive) control [45]. Thus, exercise therapy, specifically strength training, has been proven to be an effective non-surgical and non-pharmacological intervention for the effects of knee OA and is recommended in international guidelines [45]. Here, six studies regarding quadriceps and hamstring strength exercises—intended to reduce the pain in patients with knee osteoarthritis—were reviewed (Table 2). All the protocols were significantly effective in increasing quadriceps and hamstring muscle strength and decreasing pain for patients with knee OA. Most protocols were effective after at least 4 weeks of intervention, targeting quadriceps strengthening three times per week. However, the specific type of muscle-strengthening exercise and most effective session duration could not be concluded. The most effective exercise intervention programs—static quadriceps and straight leg raise exercises—revealed a statistically significant reduction in pain intensity (WOMAC scores reducing from 56.75 ± 8.43), increased range of motion (ROM), and improved function [44] after a 6-week intervention.

### 3.3. Ultrasound Therapy

Ultrasound therapy is a technique that can transform electrical energy into heat as it passes through tissues [53]. Due to its thermal and acoustic properties, it has been found that ultrasound therapy can increase pain threshold, influence neuromuscular activity to help with muscle relaxation, and help with tissue regeneration and inflammation reduction [53]. Studies investigating the treatment of knee OA with ultrasound therapy have found marked improvements in pain and joint functioning in patients who had moderate to severe knee OA [54].

Overall, the evidence indicates that all three modalities of ultrasound used in the trials (continuous, pulsed, focused low intensity) affect lowering WOMAC scores in patients (Table 3). Of the nine trials using continuous ultrasound as a treatment, eight reduced WOMAC scores by 20%. While most studies showed favorable relative risk reductions in WOMAC, only a few studies [55,56,57] yielded clinically significant relative risk reductions in stiffness WOMAC score, with −62.5%, the relative risk reduction for pain WOMAC score being the most clinically significant [57]. Discrepancies in findings can be attributed to the combination of ultrasound application with or without rigorous exercise. Of the trials that showed clinically meaningful results, the continuous ultrasound treatments used a 1 MHz frequency with an intensity of 1 W/cm^2^ or 1.5 W/cm^2^ for 5 to 10 min [38,57,58]. The pulsed ultrasound interventions similarly used a 1 MHz frequency with an intensity of 1 W/cm^2^ for 10 min at a pulsed mode of 25% [57,59]. The focused low-intensity treatment used a frequency of 0.6 MHz, a pulse repetition frequency of 300 Hz, and an average intensity of 120 mW/cm [60]. Most of the studies used 10 sessions of ultrasound treatment total, with 5 treatments per week over 2 weeks.

### 3.4. Knee Brace

Knee braces may alter the alignment of the lower extremity, decreasing the load on a specific compartment of the knee [63]. These braces are called unloader braces. Evidence suggests that unloader braces for medial knee osteoarthritis apply an external valgus force, improving the tibiofemoral alignment, shifting the body’s load away from the degenerated compartment, and reducing mechanical stress [64]. After reviewing the available studies [65,66,67,68,69], there is an overall significant decrease in pain scores after the intervention of unloader or valgus knee braces in patients with osteoarthritis (Table 4). Braces for medial knee osteoarthritis can reduce medial joint loads through three mechanisms: application of an external brace abduction moment, alteration of gait dynamics, and reduced activation of antagonistic muscles [70]. Knee braces reduced medial tibiofemoral loads primarily by applying a direct and substantial abduction moment to each subject’s knee [70]. Evidence [67,68] has found that the brace’s abduction moment reduced pain and, more particularly, that valgus bracing reduced the net varus moment about the knee by an average of 13% (7.1 N•m) and the medial compartment load at the knee by an average of 11% (114 N) in a calibrated 4° valgus brace setting [67]. For braces that deal with predominant lateral tibiofemoral OA and patellofemoral OA, the concept is similar to the unloader braces studied here. However, the number of controlled trials remains too low in the literature to prove their efficiency.

### 3.5. Electrical Stimulation

Electrical stimulation on the quadriceps muscle has been proposed to decrease muscle weakness and reduce the worsening of knee OA symptoms. Several types of electrical stimulation are currently used: high-frequency transcutaneous electrical nerve stimulation (h-TENS), low-frequency transcutaneous electrical nerve stimulation (l-TENS), neuromuscular electrical stimulation (NMES), interferential current (IFC), pulsed electrical stimulation (PES), and noninvasive interactive neurostimulation (NIN) [71]. In h-TENS, the simple application of TENS on the skin around the affected knee excites the motor neurons, facilitating movement by overriding the inhibitory mechanoreceptors signaling pain around the injured knee joint [72]. Though h-TENS was previously used for sensory relief of pain, studies have shown that h-TENS can improve motor excitability and decrease voluntary muscle activation. In IFC (considered the “gold standard” for managing knee OA widely), the stimulation works by delivering current to the skin’s deeper layers overriding the skin’s impedance [73]. As the literature shows (Table 5), IFC is the most likely to decrease pain intensity and change pain scores at last follow up [71]. Patients who use IFC have a 88% probability of showing improvement, whereas h-TENS indicates only a 74% probability. In PES, while it is comparable to h-TENS and IFC in the mechanism of action on mechanoreceptors, it differs from other forms of electrical stimulation in that it delivers current at sub-sensory intensity [74].

## 4. Data Analysis

Regarding the minimum clinically important difference for the WOMAC index for pain, we depicted in Figure 2 that only brace interventions were always above the 20% threshold, making the intervention significant and valuable for the patients regarding their Quality of Life. In addition, the mean RR for “Exercise” and “Diathermy” reached the threshold. These results indicate a higher chance for the patient to benefit from non-pharmacological intervention when the practitioner uses a brace, exercise, or diathermy.

## 5. Conclusions

This literature review helps clinicians to make evidence-based decisions for reducing knee pain and treating people living with knee OA to prevent knee replacement. This narrative review presented (i) the most recent updates on the pathogenesis of knee OA with the risk factors for developing OA and (ii) the most recent evidence for reducing knee pain with physical therapy intervention. Looking at the relative risk reduction in pain perception using the WOMAC scale for diathermy, exercise therapy, ultrasounds, knee brace, and electrical stimulation, our results show that only brace interventions always reached the minimum for clinical efficiency, which makes the intervention not only significant, but valuable for the patients regarding their quality of life. In addition, more than half of the exercise and diathermy interventions reached the minimum for clinical efficiency in reducing pain.

## Figures and Tables

**Figure 1 jcm-11-03252-f001:**
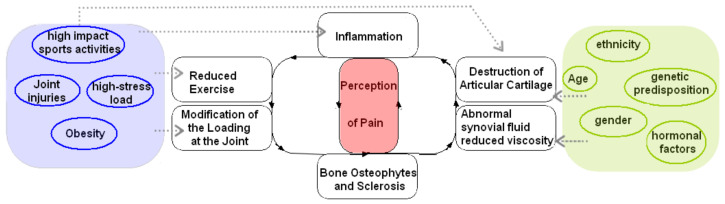
The vicious circle of OA progression and risk factors are associated where pain perception is central to the disease.

**Figure 2 jcm-11-03252-f002:**
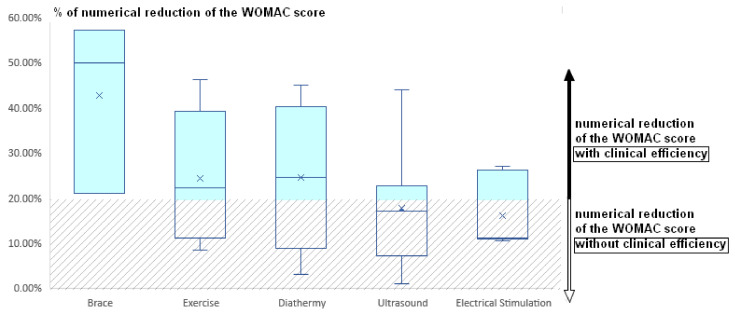
Mean percentage of numerical reduction of the WOMAC pain score from the papers with interventions regarding brace, exercise therapy, diathermy, ultrasounds, and electrical stimulation. Clinical efficiency is established when the percentage of numerical reduction is above a 20% threshold.

**Table 1 jcm-11-03252-t001:** Summary of the findings from the seven papers with interventions regarding diathermy, sample size, and the relative risk reduction for WOMAC pain subscale. H-PSWD (high-pulse short wave diathermy), L-PSWD (low-pulse short wave diathermy), MD (microwave diathermy), SWD (short-wave diathermy), SHT (superficial microwave diathermy), DHT (deep microwave diathermy), CSWD (continuous short-wave diathermy), and PSWD (pulsed short-wave diathermy).

Authors	Interventions	Sample Size	Relative Risk Reduction of Pain WOMAC
Laufer et al., (2005) [41]	H-PSWD	*n* (H-PSWD) = 32	9%
	L-PSWD	*n* (L-PSWD) = 38	3%
Giombini et al., (2011) [40]	MD	*n* (MD) = 30	44%
Rattanachaiyanont et al., (2008) [42]	SWD	*n* (SWD) = 50	27%
Rabini et al., (2012) [39]	SHT	*n* (SHT) = 27	8%
	DHT	*n* (DHT) = 27	45%
Ozen et al., (2019) [43]	CSWD	*n* (CSWD) = 14	26%
	PSWD	*n*(PSWD) = 20	23%
Boyaci et al., (2013) [38]	SWD	*n* (SWD) = 35	22%
Sarifakioglu et al., (2014) [37]	SWD	*n* (SWD) = 63	39%

**Table 2 jcm-11-03252-t002:** Summary of the findings from the seven papers with interventions regarding exercise therapy, sample size, and the relative risk reduction for WOMAC pain subscale.

Authors	Interventions	Sample Size	Relative Risk Reduction of Pain WOMAC
Vincent et al., (2019) [46]	Concentric	*n* = 28	11.3%
	Eccentric	*n* = 30	16.9%
Hall et al., (2018) [47]	Isometric	*n* = 49	8.4%
	No Intervention	*n* = 48	
Hafez et al., (2013) [48]	Pre strengthening exercises	*n* = 20	46.1%
	Post strengthening exercises		
Al-Johani et al., (2014) [49]	Conservative PT+ strengthening exercises	*n* = 20	27.7%
	*Versus* conservative PT	*n* = 20	
Oliveira et al., (2012) [50]	Instructions	*n* = 50	11%
	*Versus* quadricep strengthening	*n* = 50	
Lin et al., (2009) [51]	Strengthening exercises	*n* = 36	42.1%
	No intervention	*n* = 36	
O’Reilly et al., (1999) [52]	Strengthening exercises	*n* = 108	30.3%
	No intervention	*n* = 72	

**Table 3 jcm-11-03252-t003:** Summary of the findings from the eight papers with interventions regarding ultrasound.

Authors	Interventions	Sample Size	Relative Risk Reduction of Pain WOMAC
Boyaci et al., (2013) [38]	Continuous Ultrasound (CU)	*n* = 33	16%
	Phonophoresis (PhP)	*n* = 33	
Alfredo et al., (2020) [57]	CU	*n* = 20	6%
	Control Group (C)	*n* = 20	
Özgönenel et al., (2008) [55]	CU	*n* = 34	18%
	Sham Ultrasound Group (SU)	*n* = 33	
Luksurapan & Boonhong (2013) [53]	CU	*n* = 23	108%
	PhP	*n* = 23	
Loyola-Sánchez et al., (2012) [59]	Pulsed Ultrasound Group (PU)	*n* = 14	23%
	Sham Ultrasound Group (SU)	*n* = 13	
Kozanoglu et al., (2003) [61]	CU	*n* = 30	22%
	Ibuprofen Phonophoresis (PH)	*n* = 30	
Külcü et al., (2009) [62]	CU	*n* = 15	44%
	SU	*n* = 15	
Karakaş et al., (2020 [56])	CU	*n* = 39	11%
	C	*n* = 36	

**Table 4 jcm-11-03252-t004:** Summary of the findings from the five papers with interventions regarding knee brace.

Authors	Interventions	Sample Size	Relative Risk Reduction of Pain WOMAC
Hurley et al., (2012) [65]	Valgus Unloader Knee Brace	*n* = 24	21%
Briggs et al., (2012) [66]	Valgus Unloader Knee Brace	*n* = 39	57.1%
Pollo et al., (2002) [67]	Valgus Unloader Knee Brace	*n* = 11	44.4% (VAS)
Fatani-Pagani et al., (2010) [68]	Valgus Unloader Knee Brace	*n* = 11	50%%
Richards et al., (2005) [69]	Valgus Unloader Knee Brace	*n* = 30	41% (VAS)

**Table 5 jcm-11-03252-t005:** Summary of the findings from the six papers with interventions regarding electrical stimulation.

Authors	Interventions	Sample Size	Relative Risk Reduction of Pain WOMAC
Atamaz et al., (2012) [73]	TENS	*n* = 29	11%
	IFC	*n* = 27	11%
Pietrosimone et al., (2020) [72]	TENS + TE	*n* = 30	10.57%
	sham TENS + TE	*n* = 30	3.3%
	TE only	*n* = 30	12%
Adedoyin et al., (2005) [75]	exercise+ electrical stimulation	*n* = 16	27%
	exercise	*n* = 11	7%
Garland et al., (2007) [76]	Active device	*n* = 38	26%%
	Placebo device	*n* = 20	7%
Fary et al., (2011) [74]	Pulse Electrical Stimulation	*n* = 34	11%
	placebo	*n* = 36	
Shimoura et al., (2019) [77]	TENS Stair climb	*n* = 50	33% (VAS)
	TENS Timed up and go		26% (VAS)
	TENS 6 mi walk test		55% (VAS)

## Data Availability

Not applicable.

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
