# Peer review of "Knee Pain from Osteoarthritis: Pathogenesis, Risk Factors, and Recent Evidence on Physical Therapy Interventions"

_jcm, 2022, doi:10.3390/jcm11123252_

Round 1
Reviewer 1 Report
Great review. I have one question regarding knee braces: the authors only reviewed unloader braces for predominant tibiofemoral OA, how about braces for predominant lateral tibiofemoral OA and patellofemoral OA?
Author Response
R1
Great review.
I have one question regarding knee braces: the authors only reviewed unloader braces for predominant tibiofemoral OA; how about braces for predominant lateral tibiofemoral OA and patellofemoral OA?
We thank reviewer 1 for their strong support, and we are happy to elaborate on the comments provided.
Knee braces for knee arthritis are either unicompartmental offloaders or tri-compartment offloaders. Uni-compartment offloaders work by redistributing pressure from one side of the knee joint to the other side. Tri-compartment offloaders work by reducing pressure across the entire joint. Since the literature remains limited on multicompartment and mainly focused on tibiofemoral OA, we wanted to propose an analysis that we could compare with the other techniques. Similarly, we only found two papers dealing with Patellofemoral OA (see below) that cannot help us to conclude.
Thus, to consider R1's comments, we mentioned that other braces existed, but the level of evidence remains low to be integrated into this review.
We changed the text as follows:
“For braces that deal with predominant lateral tibiofemoral OA and patellofemoral OA, the concept is similar to the unloader braces studied here. However, the number of controlled trials remains too low in the literature to prove their efficiency.”
Hunter DJ, Harvey W, Gross KD, et al. A randomized trial of patellofemoral bracing for treatment of patellofemoral Osteoarthritis. Osteoarthritis Cartilage 2011;19:792–800.
Callaghan MJ, Parkes MJ, Hutchinson CE, et al. A randomized trial of a brace for patellofemoral Osteoarthritis targeting knee pain and bone marrow lesions Annals of the Rheumatic Diseases 2015;74:1164-1170.
Reviewer 2 Report
From my point of view, the work is well-done and provides new non-pharmacological conservative treatments in Orthopedics and Rehabilitation to prevent Osteoarthritis (OA) progression and avoid knee surgery. This review can help clinicians to make evidence-based decisions for reducing knee pain and treating people living with knee OA to prevent knee replacement. Just, I suggest some minor language modifications.
Author Response
R2
From my point of view, the work is well-done and provides new non-pharmacological conservative treatments in Orthopedics and Rehabilitation
to prevent Osteoarthritis (OA) progression and avoid knee surgery.
This review can help clinicians to make evidence-based decisions for reducing knee pain and treating people living with knee OA to prevent knee replacement.
Just, I suggest some minor language modifications.
We thank reviewer 2 for their strong support, and we are happy to elaborate on the comments provided. We agree that we could have better reviewed the paper before submission. The new version has been reviewed thoroughly by an external reviewer to correct typos and unclear sentences.
Reviewer 3 Report
The author investigated the issues associated with knee pain in OA and the importance of physical therapy in pain prevention.
Comments
1. The paper requires improvements in scientific writing.
2. Lines 119-121: obesity contributes to OA not only due to increased loading on the joints. This should be corrected.
3. Lines 49-51: Hypertrophy development is not solely associated with aging. This should be corrected.
4. Line 53: MMP1, MMP3 and HSPA1A do not belong to chemokines. This should be corrected.
5. Lines 54-56: Cartilage degradation results primarily from the destruction of the extracellular matrix. This should be corrected.
6. Line 75: It is not clear what the author means by” changes in synovitis”. This should be corrected.
7. Lines 150-151: The end of the sentence is not clear. This should be clarified.
8. Lines 202-203: It is not clear what the author means by “to reduce synovial thickness”. This should be clarified.
Author Response
R3
The author investigated the issues associated with knee pain in OA and the importance of physical therapy in pain prevention.
We thank reviewer 3 for their strong support, and we are happy to elaborate on the comments provided.
Comments
- The paper requires improvements in scientific writing.
We agree that we could have better reviewed the paper before submission. The new version has been reviewed thoroughly by an external reviewer to correct typos and unclear sentences.
- Lines 119-121: Obesity contributes to OA not only due to increased loading on the joints. This should be corrected.
The author agreed with Reviewer 3 and decided to include the metabolic and inflammatory factors that relate OA to Obesity.
We changed our text such as:
“Obesity increases the risk of developing OA through systemic and biomechanical factors. For instance, Obesity alters (i) metabolism and joint inflammation that contributes to OA in non-weight-bearing joints such as the hands and (ii) biomechanical loading on weight-bearing joints such as the knee or the hip. Regarding the biomechanical factors - based on the multiplier effect of lever arms outside the body's central axis –, a force of three to six times the body weight is exerted across the knee during a single-leg stance in walking”
- Lines 49-51: Hypertrophy development is not solely associated with aging. This should be corrected.
The author agreed with Reviewer 3 and decided to include Oxidative stress in addition to aging.
We changed our text such as:
“For instance, cartilage cells’ (i.e., chondrocytes') compositional and structural alterations - like hypertrophy due to aging or oxidative stress - trigger the production of catabolic factors enhancing cartilage debilitation.”
- Line 53: MMP1, MMP3, and HSPA1A do not belong to chemokines. This should be corrected.
The author agreed with Reviewer 3 and decided to correct the labeling.
We changed our text such as:
“– cytokines (e.g., IL-6, IL-8), chemokines (e.g., RANTES, IP-10), metalloproteases (MMP1, MMP3), and heat shock proteins (e.g., HSPA1A) –"
- Lines 54-56: Cartilage degradation results primarily from the destruction of the extracellular matrix. This should be corrected.
The author agreed with Reviewer 3 and decided to correct the text.
We changed our text such as:
“cartilage degradation coming from the destruction of the extracellular matrix”
- Line 75: It is not clear what the author means by" changes in synovitis." This should be corrected.
Thanks to Reviewer 3 comments, we clarified our wording such as :
"Previous studies showed pain has been associated with several structural factors including bone marrow lesions, synovial thickening (synovitis), and knee effusion [9]."
- Lines 150-151: The end of the sentence is not clear. This should be clarified.
To avoid any confusion, we rephrased the paragraph such as:
"Regarding joint malalignment - changes in joint geometry- it decreases the joint's ability to adapt to its biomechanical environment, contributing to cartilage or bone tissue damage. For instance, Varus knee malalignment and dynamic knee adduction moments have been found to cause medial compartment knee OA due to the increase in mechanical stress on the medial compartment of the knee; the reverse is valid for a valgus knee align-ment[29–31]."
- Lines 202-203: It is not clear what the author means by "to reduce synovial thickness." This should be clarified.
Synovial thickness is a prognostic marker of cartilage loss. Therefore, the synovial thickness may be regarded as an index in assessing the extent of the synovitis. The use of diathermy has been considered successful when synovial thickness has been reduced.
We clarified our sentence as follows:
“Deep microwave diathermy has been shown to reduce synovial thickness- a prognostic marker of cartilage loss - in patients with knee OA, which assists in decreasing pain associated with synovitis and the progression of cartilage loss."
Reviewer 4 Report
Title: good
Abstract: good, but it is implicated that knee osteoarthritis could be prevented by training/exercise. This wording has to be weakened.
Introduction: Gives a good overview
Main text and analysis: well described and adequate
Conclusion: Coherent and message well transported.
References: All major and important literature is cited
Dear Authors,
Thanks for submitting your Manuscript to JCM. The background of and physical therapy for Osteoarthritis is reviewed. The most important literature was appreciated.
The manuscript methodology is adequate and it is well written.
Please be careful and weaken your wording regarding the prevention of osteoarthritis. We don't now, if we can prevent OA by exercise. Probably not.
Author Response
R4
Title: good
Abstract: good, but it is implicated that knee osteoarthritis could be prevented by training/exercise. This wording has to be weakened.
To avoid confusion, we changed our wording such as:
“In addition, more than half of the Exercise and Diathermy interventions reached the minimum for clinical efficiency regarding pain level."
Introduction: Gives a good overview
Main text and analysis: well described and adequate
Conclusion: Coherent and message well transported.
References: All major and important literature is cited
Dear Authors,
Thanks for submitting your Manuscript to JCM. The background of and physical therapy for Osteoarthritis is reviewed. The most important literature was appreciated.
The manuscript methodology is adequate, and it is well written.
Please be careful and weaken your wording regarding the prevention of Osteoarthritis. We don't know if we can prevent OA by exercise. Probably not.
We thank reviewer 4 for their strong support, and we are happy to elaborate on the comments provided.
To avoid confusion, we changed our wording such as:
“In addition, more than half of the Exercise and Diathermy interventions reached the minimum for clinical efficiency regarding pain level."